# AlGaN Quantum Disk Nanorods with Efficient UV-B Emission Grown on Si(111) Using Molecular Beam Epitaxy

**DOI:** 10.3390/nano12142508

**Published:** 2022-07-21

**Authors:** Dongqi Zhang, Tao Tao, Haiding Sun, Siqi Li, Hongfeng Jia, Huabin Yu, Pengfei Shao, Zhenhua Li, Yaozheng Wu, Zili Xie, Ke Wang, Shibing Long, Bin Liu, Rong Zhang, Youdou Zheng

**Affiliations:** 1Key Laboratory of Advanced Photonic and Electronic Materials, School of Electronic Science and Engineering, Nanjing University, Nanjing 210023, China; dz1923016@smail.nju.edu.cn (D.Z.); lsqnino@163.com (S.L.); suiyunshao@sina.com (P.S.); lzh325a@sina.com (Z.L.); 18251900811@163.com (Y.W.); xzl@nju.edu.cn (Z.X.); kewang@nju.edu.cn (K.W.); rzhangxmu@xmu.edu.cn (R.Z.); ydzheng@nju.edu.cn (Y.Z.); 2School of Microelectronics, University of Science and Technology of China, Hefei 230026, China; xjnj@mail.ustc.edu.cn (H.J.); huabin@mail.ustc.edu.cn (H.Y.); shibinglong@ustc.edu.cn (S.L.); 3Institute of Future Display Technology, Tan Kah Kee Innovation Laboratory, Xiamen 361102, China

**Keywords:** molecular beam epitaxy, nanorods, AlGaN, quantum disks

## Abstract

AlGaN nanorods have attracted increasing amounts of attention for use in ultraviolet (UV) optoelectronic devices. Here, self-assembled AlGaN nanorods with embedding quantum disks (Qdisks) were grown on Si(111) using plasma-assisted molecular beam epitaxy (PA-MBE). The morphology and quantum construction of the nanorods were investigated and well-oriented and nearly defect-free nanorods were shown to have a high density of about 2 × 10^10^ cm^−2^. By controlling the substrate temperature and Al/Ga ratio, the emission wavelengths of the nanorods could be adjusted from 276 nm to 330 nm. By optimizing the structures and growth parameters of the Qdisks, a high internal quantum efficiency (IQE) of the AlGaN Qdisk nanorods of up to 77% was obtained at 305 nm, which also exhibited a shift in the small emission wavelength peak with respect to the increasing temperatures during the PL measurements.

## 1. Introduction

Over the past decade, much attention has been paid to ultraviolet (UV) optoelectronic devices due to their wide applications in the food industry, biological detection, medical treatment, and integrated circuit fields [1,2,3,4,5,6]. AlGaN-based ultraviolet LED devices are believed to be one of the best choices because they are more environmentally friendly and potentially have a higher efficiency than conventional UV lamps [7,8]. Therefore, AlGaN ternary materials have been widely investigated for use in optical devices, such as UV light-emitting diodes (LEDs), UV lasers (LDs), etc. [9,10]. However, compared to the well-established GaN-based blue LED devices, AlGaN-based optical devices still present many challenges that need to be solved, such as low external quantum efficiency (EQE) (especially in the deep UV range), high dislocation densities, low light extraction efficiency (LEE), and non-efficient p-doping [11,12,13]. To date, the highest reported EQE of an AlGaN-based LED device was only 20.3%, which is much lower than that of GaN-based blue LED devices [14].

Recently, AlGaN nanorods have emerged as an alternative to high-performance UV emitter devices [15]. Compared to AlGaN film structures, the nanorod structures have many advantages [16,17,18,19]. The bottom-up growth mode produces nanorods with low dislocation densities and high crystalline quality as the most commonly reported major advantages, which are mainly attributed to efficient lateral stress relaxation of the nanorod structures. In addition, Mg-doping is significantly enhanced in nanorod structures compared to films because the formation energy of the group-III metal substitutional Mg-dopant has been found to be significantly reduced in the near-surface region of nanorod structures [20,21]. It should also be mentioned that nanorods can be grown on many different substrates. There are many reports of nanorods being grown on Si, sapphire, SiO_2_, metal, etc. [17,22,23,24]. In other words, nanorod structures could break through the substrate limits that are caused by the large lattice mismatch between the epitaxial layer and the substrate to realize nanorod devices with good performances. 

Many efforts have been made to develop AlGaN-based nanorods. Daudin et al. conducted a deep investigation into the growth techniques and structural properties of AlGaN nanorods. Recently, they used extended diffraction anomalous fine structures (EDAFSs) to reveal the short-range ordering of Al and Ga atoms and the composition fluctuations in AlGaN nanorods. It was found that the local and average compositions of AlGaN nanorods are different depending on the growth parameters [25,26]. Mi et al. reported AlGaN nanorods devices with high efficiency and deep UV emission, in which high-quality AlGaN nanorod heterostructures and efficient p-type conduction AlN nanorods were demonstrated [21,27,28,29]. Based on that, electrically injected nanorod lasers that operated at 239 nm at room temperature were developed with a threshold current of 0.35 mA [30]. 

In this work, AlGaN-based nanorods with embedding quantum disks (Qdisks) were designed and grown using plasma-assisted molecular beam epitaxy (PA-MBE) on an Si(111) substrate. The morphology of the GaN nanorods and the atomic construction of the AlGaN Qdisk nanorods were investigated using SEM and HR-TEM. By controlling the substrate temperature and Al/Ga ratio, the emission wavelengths of the AlGaN Qdisk nanorods could be adjusted from 276 nm to 330 nm. Finally, after systematic optimization, the internal quantum efficiency (IQE) of the AlGaN Qdisk nanorods was enhanced to as much as 77%, in which a shift in the small wavelength peak with the temperature change indicated a lower QCSE.

## 2. Materials and Methods

### 2.1. Fabrication of the AlGaN Qdisk Nanorods

A Riber Compact 21 plasma-assisted molecular beam epitaxy system that was equipped with a CESAR PF power generator was employed to grow the nanorod structures on 2-inch Si(111) substrates. Before the process started, all substrates were cleaned in acetone, methanol, hydrochloric acid, and deionized water to remove surface contaminants and were then etched in buffered oxide etch (BOE) for 15 min to remove any oxides on their surfaces. Then, the substrates were outgassed at 400 ℃ in a buffer chamber for 30 min. Subsequently, the substrates were thermally cleaned at 840 ℃ in the growth chamber to further remove any residual oxides. After 15 min of heat treatment, 7 × 7 surface reconstruction RHEED patterns appeared. The temperature that is mentioned here refers to the thermocouple reading on the backside of the substrate. 

A schematic of the AlGaN Qdisk nanorods is shown in Figure 1a and a schematic diagram of the growth process is shown in Figure 1b. The process was as follows: (Ⅰ) an AlN-seeding layer was deposited; (Ⅱ)vertically aligned GaN nanorods that were about 200 nm long were spontaneously formed at 780 ℃ [31]; (Ⅲ) AlGaN nanorod segments that were about 40 nm in length were deposited; (Ⅳ) by alternating the two Al cells, five pairs of Al_x_Ga_1−x_N (2.5nm) / Al_y_Ga_1−y_N (5nm) (x < y) Qdisks were grown as the active regions; (Ⅴ) the final 40-nm AlGaN nanorod segments were deposited. The Ga flux was maintained at 1 × 10^−7^ Torr during the growth procedure and the Al flux varied from 3 × 10^−8^ Torr to 8 × 10^−8^ Torr. To facilitate the growth of the AlGaN nanorods on the Si substrates, a GaN nanorod template was grown previously, which performed as a shield that prevented the Al atoms from reaching the Si substrates; otherwise, highly coalesced nanorods and quasi-film-like structures would form because of the low mobility of the Al atoms. To investigate the influence of the Al atoms on the growth of the nanorods, samples (a)–(c) were grown. The substrate temperate was set at 780 ℃ and the Ga flux and Al flux were 1 × 10^−7^ Torr and 5 × 10^−8^ Torr, respectively. Sample (a) comprised AlGaN nanorods that were grown directly on the Si substrates. Sample (b) comprised AlGaN nanorods that were grown on Si substrates but started with GaN nanorods. Sample (c) comprised AlGaN nanorods that were grown on Si substrates and started with GaN nanorods, but the Al flux was increased to 1 × 10^−7^ Torr. In addition, by controlling the Al flux between 2 × 10^−8^ Torr and 8 × 10^−8^ Torr, another series of samples (d)–(i) with emission wavelengths from 330 nm to 276 nm was obtained. 

### 2.2. Characterization

The morphology of the nanorods was captured using a JEOL JSF-7000F field emission scanning electron microscope (FE-SEM). The atomic structures of the AlGaN nanorods were examined using a Titan scanning transmission electron microscope (STEM) and a TALOS 200 high-resolution transmission electron microscope (HR-TEM). The PL spectra of the nanorods were excited using a 266-nm PSU-H-FDA frequency multiplier laser and collected using an IsoPlane SCT 320 optical spectrometer.

## 3. Results and Discussion

The growth process of the nanorods was monitored in situ using reflective high-energy electron diffraction (RHEED). The SEM images of samples (a)–(c) are shown in Figure 2a–c and the corresponding RHEED images are shown in Figure 2d–f, respectively. The RHEED image of sample (a) showed bright spots in good alignment; however, the structures that are shown in Figure 2d could not be called nanorods. Actually, they were quasi-film-like structures due to the low mobility of the Al atoms [32]. The structures in Figure 2d were obtained by growing the AlGaN nanorods directly on the Si substrates; meanwhile, the nanorod structures in Figure 2e were obtained by growing AlGaN nanorods on the GaN nanorods. The GaN nanorods played an important role in preventing the Al atoms from reaching the Si substrates during the growth process. By introducing the GaN NWs, well-oriented and independent nanorods could be obtained. Figure 2b shows the RHEED image of sample (b). With the GaN nanorods that were grown prior, the RHEED pattern consisted of bright fine spots. As can be seen in the corresponding SEM image (Figure 2e), well-oriented nanorods were formed. The density of the nanorods was about 2 × 10^10^ cm^−2^ and the height of the nanorods was about 300 nm. When the Al flux increased to 1 × 10^−7^ Torr for sample (c), the nanorods were close together and started to coalesce. The RHEED image could be divided into spots and lines. The spots indicated the AlGaN nanorods and the weak lines showed similar characteristics to AlGaN film. As can be seen in the SEM image, the tops of the AlGaN nanorods were flat and the spaces between the AlGaN nanorods were smaller than those between the AlGaN nanorods in sample (b). As the Al atom flux increased, the nanorods had the tendency to coalesce and become film because of the increased lateral growth rate, which resulted from the slow migration rate of the Al atoms. Our results agreed with reports of AlGaN film being formed by the coalescence of nanorods [33].

The morphology and atomic structures of the AlGaN Qdisk nanorods were examined using STEM and the results are shown in Figure 3. One independent AlGaN nanorod is illustrated in Figure 3a, which could be divided into four parts. Region Ⅰ was the GaN nanorod template, which was approximately 200 nm in length. Region Ⅱ was an Al-rich AlGaN region between the GaN and AlGaN nanorods. When the GaN nanorods were exposed to Al and Ga atoms simultaneously, Al-rich AlGaN was more favorable for nucleation due to the higher stability of the Al–N bond than that of the Ga–N bond, thus causing the presence of the Al-rich AlGaN region II. Then, the Ga atoms accumulated to a certain level, transforming the Al-rich AlGaN region into a stable AlGaN growth region (region III) [26]. The next two regions, Ⅲ and Ⅳ, were the undoped AlGaN region and the active AlGaN/GaN Qdisk region, respectively. Our elemental analysis using energy dispersive x-ray (EDX) further confirmed the four separate regions, as shown by the Al and Ga distributions that were scanned along the line a–b in Figure 3b. Regions Ⅰ, Ⅱ, and Ⅲ and the Qdisk region Ⅳ could be clearly distinguished. The variations in the Al and Ga components met the characteristics of each region.

Figure 3c shows a HR-TEM image of the active Qdisk region. Highly ordered atoms could be seen, which indicated a homogeneous component distribution. Additionally, it was found that the nanorods were nearly defect-free during our observations, further proving the high crystalline quality of the AlGaN nanorods. Figure 3d,e show amplified TEM images of the well-defined uniform active region, including five Al_x_Ga_1−x_N Qdisks that were separated by Al_y_Ga_1−y_N barrier layers (x < y). The active region was about 32 nm in length and consisted of five pairs of 2.5-nm Qdisks and 5-nm layers. The Qdisks and layers were clearly distinguished by the different contrasts that were introduced by the different Al components levels. The clear boundaries between the Qdisks and the barriers indicated the fine uniformity of the Al composition. Figure 3d shows that there were bulges at the edges of the quantum disks, which were the result of a kinetic process during the nanorod growth procedure [34].

A series of AlGaN samples (d)–(i) with different emission peaks were demonstrated by adjusting the Al flux during the growth process. The Ga flux was fixed at 1 × 10^−7^ Torr and the temperature was set at 810 ℃ during the growth of the AlGaN nanorods. The Al flux for the growth of the Qdisks varied from 2 × 10^−8^ Torr to 8 × 10^−8^ Torr. The structures that can be seen in Figure 3 were the typical structures of the nanorods in samples (d)–(i). The structures of the nanorods in samples (d)–(i) were similar, including the GaN template, the AlGaN segments and the five pairs of Al_x_Ga_1−x_N (2~3 nm) / Al_y_Ga_1−y_N (5~6 nm) (x < y) Qdisks. Therefore, the wavelength change could be attributed to the Al composition change in the quantum disks and barriers. The PL spectra of samples (d)–(i) were measured at room temperature and are shown in Figure 4a. The emission wavelengths of samples (d)–(i) varied from 330 nm to 276 nm. The average full width at half maximum (FWHM) of the PL spectra was about 25 nm. As the wavelength varied, no significant changes in the FWHM were observed. Wavelengths as low as 276 nm were obtained when the Al flux was at 8 × 10^−8^ Torr. 

The temperature-dependent photoluminescence (TDPL) spectra of the AlGaN Qdisk nanorods were adopted to explore the optical properties of the nanorods [10]. The TDPL spectra were measured using a 266 laser as the exciter. Figure 4b illustrates the TDPL spectra of sample (f), which were measured at temperatures ranging from 10 K to 300 K. Two peaks can be seen at around 306 nm and 355 nm, which corresponded to the emissions of the AlGaN and GaN nanorods. By assuming that the IQE at 10 K was 100%, the IQE could be estimated as about 77% using the ratio of the integral PL intensity at 300 K to that at 10 K. The relative IQE values and wavelength shifts of the AlGaN nanorods are shown in Figure 4c with respect to temperature. The time-resolved photoluminescence (TRPL) of sample (f) at 300 K is shown in the inset image. The TRPL spectra could be fitted using a biexponential function:(1)$$I\left(t\right)=A_{1}\exp\left(-\frac{t}{\tau_{1}}\right)+A_{2}\exp\left(-\frac{t}{\tau_{2}}\right)$$
where $I\left(t\right)$ is the luminescence intensity, $A_{1}$ and $\tau_{1}$ are the fast decay components, and $A_{2}$ and $\tau_{2}$ are the slow decay components. According to the fitting results, $\tau_{1}$ was 0.09 ns and $\tau_{2}$ was 4.73 ns. 

The emissions of the AlGaN nanorods in sample (f) exhibited a small redshift (~3.5 nm) as the temperature increased to 300 K. The redshift could be explained by the energy band variations with respect to the temperature. Normally, the energy bandgap of semiconductor materials is variable with temperature, which can be described by the Varshni empirical formula [35]:(2)$$E_{g}\left(T\right)=E_{g}\left(0\right)-\frac{\alpha T^{2}}{\beta +T}$$
where *E_g_*$\left(0\right)$ is the material energy band at the temperature of 0 K, *α* is the match constant for the material, and *β* is a parameter that is connected to the Debye temperature. For a GaN-based semiconductor material, the values of *α* and *β* could be as assumed to be constant. According to the Varshni empirical formula, the emission peak shift of the AlGaN nanorods could be fitted and is shown in Figure 4d. The fitted *E_g_*$\left(0\right)$ was 4.101 eV and the gap between the fitted curve and the experimental data at 10 K was 10 meV, which could be related to some localized states in the AlGaN Qdisk nanorods. At low temperatures, local exciton emissions that were caused by alloy fluctuations dominated the PL peak energy and the emission peak shift was insignificant below the temperature of 150 K. As the temperature increased, the excitons were thermally activated and escaped from the localized states. With the temperature increase, the PL peak wavelength redshifted due to the bandgap variations and followed the Varshni equation [36].

## 4. Conclusions

In summary, AlGaN quantum disk nanorods that were grown on Si(111) substrates were investigated by carefully controlling the MBE growth parameters. The SEM and TEM measurements exhibited well-oriented and nearly defect-free AlGaN nanorods. The quantum disks were clearly defined, indicating the sharp interface and fine uniformity of the Al composition. By adjusting the Al/Ga ratio and substrate temperature, the emission wavelengths of the AlGaN nanorods could be adjusted from 276 nm to 330 nm. The optimized nanorods showed high IQE values of up to 77% and a 3.5-nm emission peak shift that was proven by the TDPL measurements, which indicated that the Qdisks were well designed and had a good crystalline quality. These MBE-grown AlGaN nanorods are expected to have great potential to develop high performance UV emitters.

## Figures and Tables

**Figure 1 nanomaterials-12-02508-f001:**
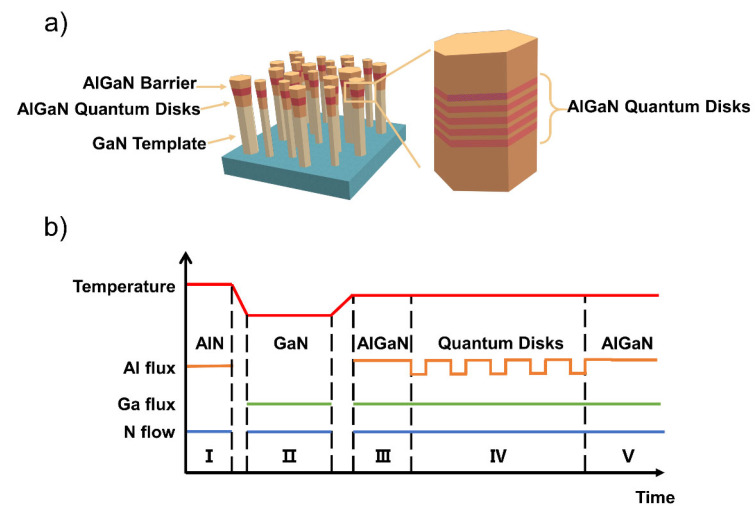
(**a**) The schematic of the AlGaN Qdisk nanorods; (**b**) the schematic diagram of the growth process.

**Figure 2 nanomaterials-12-02508-f002:**
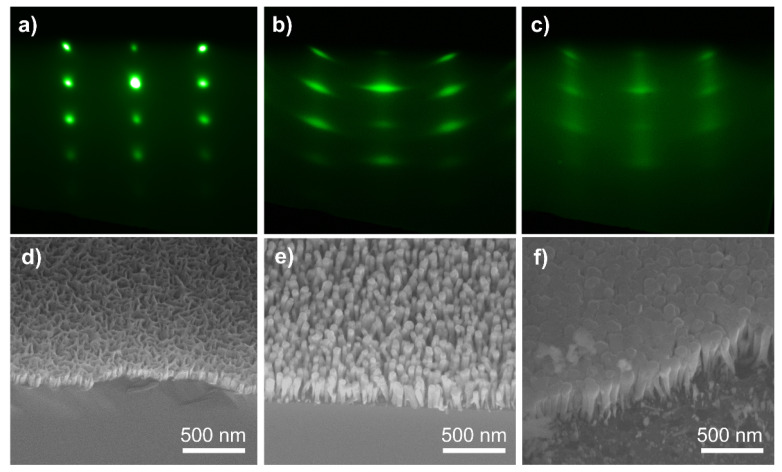
(**a**–**c**) the RHEED images of the AlGaN nanowires samples (**a**–**c**); (**d**–**f**) the SEM images of the AlGaN nanowires samples (**a**–**c**).

**Figure 3 nanomaterials-12-02508-f003:**
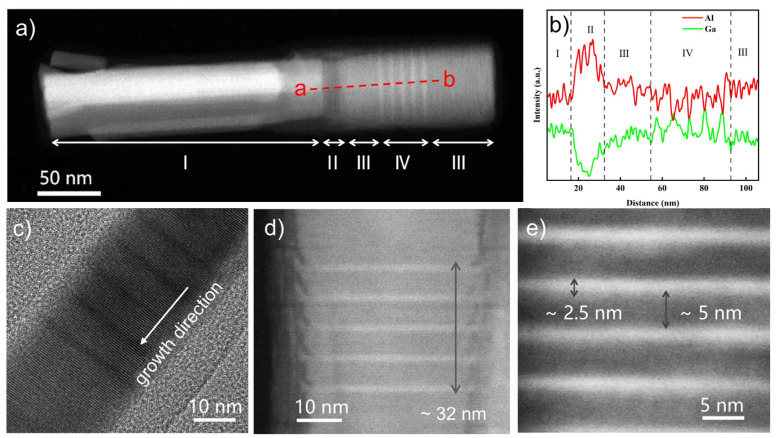
(**a**) STEM image of a single AlGaN nanowire, (**b**) EDX image along the growth direction, (**c**) HR-TEM images of the Qdisks region. (**d,e**) STEM images of the Qdisks region.

**Figure 4 nanomaterials-12-02508-f004:**
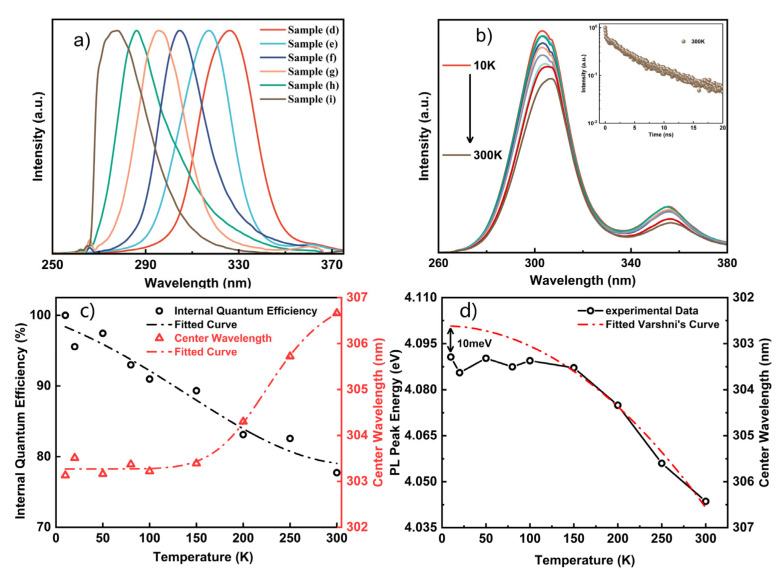
(**a**) PL spectra of the AlGaN nanowires with different emission wavelengths, (**b**) TDPL spectra of the AlGaN nanowires emission at 305 nm, (**c**) IQE and wavelength change curves with temperature change, (**d**) the wavelength shifts fitting curve using the Varshni empirical formula.

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
