# Peer review of "AlGaN Quantum Disk Nanorods with Efficient UV-B Emission Grown on Si(111) Using Molecular Beam Epitaxy"

_nanomaterials, 2022, doi:10.3390/nano12142508_

Round 1

Reviewer 1 Report

Regarding Fig. 2a '- the structures shown in this picture can hardly be called nanowires.

"It is interesting that the emission peak shift is insignificant below the temperature of 150 K." Do the authors have any hypothesis why, below the temperature of 150K, the shift of the emission peak is practically not visible? Referring to Fig. 4d, it seems that between 150K and 300K the peak emission shift can be fitted with a straight line! What do the authors think about this?

Author Response

Thank you for your questions concerning our manuscript entitled “AlGaN quantum disks nanowires with efficient UV-B emission grown on Si (111) by molecular beam epitaxy” (ID:1799192). Those questions are very valuable and helpful for revising and improving our paper, as well as the important guiding significance to our research. We have carefully made revision according to those questions, which we hope to meet with approval. Revised portions are marked in  “Track Changes”  in the manuscript.  The responses are upload and please see the attachment.  

Reviewer 2 Report

The manuscript presents results of investigation of morphology and luminescence of AlGaN-based nanowires with embedding quantum disks (Qdisks) grown by plasma-assisted molecular beam epitaxy (PA-MBE) on the Si (111)  substrate. It is shown that the emission wavelength of AlGaN Qdisks nanowires can be adjusted from 276 nm to 330 nm  by carefully controlling the substrate temperature and Al/Ga ratio. The obtained results are very interesting and practically useful.

The paper can be published.

Author Response

Thank you very much for reviewing the manuscript and your very encouraging comments on the merits.

Reviewer 3 Report

The manuscript by Zhang et al. reports on the fabrication and structural/optical characterization of AlGaN quantum disk nanowires grown on silicon using plasma-assisted MBE. The authors realize high-quality and high-density films of nanowires with embedded AlGaN quantum disks, the growth of which they control using the Al flow. The wires are characterized structurally using STEM and their photoluminescence spectra in the UV (250-370 nm) are measured as a function of temperature, displaying high internal quantum efficiency and variable central emission frequency as a function of the disks' structure.

The results are relevant for the realization of efficient UV emitting nanodevices and, as such, fits well within the topics covered by a journal such as Nanomaterials. The fabrication and characterization of the devices are in general clearly presented and the manuscript could thus be publishable in this journal.

My only concern is the correlation between the observed emission shifts of Fig. 4a and the changes in the disks' structure following the variation of the Al flow during deposition. The authors very succintly introduce the samples (d)-(i), but quantitative informations on their structure, as those given in Fig. 3, are missing. It is also not clear if the authors can, based on such a structural analysis, account for the 3.75 to 4.49 eV change in bandgap or if these numbers are just based on the observed PL shifts. To summarize, the paper would be strengthened by a clearer structural characterization of the different nanowires as well as a more quantitative discussion of the observed shifts.

Author Response

(The authors gave the same response as above.)

Round 2

Reviewer 3 Report

The authors have reasonably addressed my previous comments. I can recommend publication in Nanomaterials.

Author Response

Thank you very much for your very encouraging comments on the manuscript.